# Scaling provable adversarial defenses

**Eric Wong**
Machine Learning Department
Carnegie Mellon University
Pittsburgh, PA 15213
`ericwong@cs.cmu.edu`

**Frank R. Schmidt**
Bosch Center for Artificial Intelligence
Renningen, Germany
`frank.r.schmidt@de.bosch.com`

**Jan Hendrik Metzen**
Bosch Center for Artificial Intelligence
Renningen, Germany
`janhendrik.metzen@de.bosch.com`

**J. Zico Kolter**
Computer Science Department
Carnegie Mellon University and
Bosch Center for Artificial Intelligence
Pittsburgh, PA 15213
`zkolter@cs.cmu.edu`

## Abstract

Recent work has developed methods for learning deep network classifiers that are *provably* robust to norm-bounded adversarial perturbation; however, these methods are currently only possible for relatively small feedforward networks. In this paper, in an effort to scale these approaches to substantially larger models, we extend previous work in three main directions. First, we present a technique for extending these training procedures to much more general networks, with skip connections (such as ResNets) and general nonlinearities; the approach is fully modular, and can be implemented automatically (analogous to automatic differentiation). Second, in the specific case of $\ell_\infty$ adversarial perturbations and networks with ReLU nonlinearities, we adopt a nonlinear random projection for training, which scales *linearly* in the number of hidden units (previous approaches scaled quadratically). Third, we show how to further improve robust error through cascade models. On both MNIST and CIFAR data sets, we train classifiers that improve substantially on the state of the art in provable robust adversarial error bounds: from 5.8% to 3.1% on MNIST (with $\ell_\infty$ perturbations of $\epsilon = 0.1$), and from 80% to 36.4% on CIFAR (with $\ell_\infty$ perturbations of $\epsilon = 2/255$). Code for all experiments in the paper is available at `https://github.com/locuslab/convex_adversarial/`.

## 1 Introduction

A body of recent work in adversarial machine learning has shown that it is possible to learn *provably robust* deep classifiers [Wong and Kolter, 2017, Raghunathan et al., 2018, Dvijotham et al., 2018]. These are deep networks that are verifiably *guaranteed* to be robust to adversarial perturbations under some specified attack model; for example, a certain robustness certificate may guarantee that for a given example $x$, no perturbation $\Delta$ with $\ell_\infty$ norm less than some specified $\epsilon$ could change the class label that the network predicts for the perturbed example $x + \Delta$. However, up until this point, such provable guarantees have only been possible for reasonably small-sized networks. It has remained unclear whether these methods could extend to larger, more representionally complex networks.

In this paper, we make substantial progress towards the goal of scaling these provably robust networks to realistic sizes. Specifically, we extend the techniques of Wong and Kolter [2017] in three key ways. First, while past work has only applied to pure feedforward networks, we extend the framework to deal with arbitrary residual/skip connections (a hallmark of modern deep network architectures),

and arbitrary activation functions (Dvijotham et al. [2018] also worked with arbitrary activation functions, but only for feedforward networks, and just discusses network verification rather than robust training). Second, and possibly most importantly, computing the upper bound on the robust loss in [Wong and Kolter, 2017] in the worst case scales *quadratically* in the number of hidden units in the network, making the approach impractical for larger networks. In this work, we use a nonlinear random projection technique to estimate the bound in manner that scales only linearly in the size of the hidden units (i.e., only a constant multiple times the cost of traditional training), and which empirically can be used to train the networks with no degradation in performance from the previous work. Third, we show how to further improve robust performance of these methods, though at the expense of worse non-robust error, using multi-stage cascade models. Through these extensions, we are able to improve substantially upon the verified robust errors obtained by past work.

## 2   Background and related work

Work in adversarial defenses typically falls in one of three primary categories. First, there is ongoing work in developing heuristic defenses against adversarial examples: [Goodfellow et al., 2015, Papernot et al., 2016, Kurakin et al., 2017, Metzen et al., 2017] to name a few. While this work is largely empirical at this point, substantial progress has been made towards developing networks that seem much more robust than previous approaches. Although a distressingly large number of these defenses are quickly "broken" by more advanced attacks [Athalye et al., 2018], there have also been some methods that have proven empirically resistant to the current suite of attacks; the recent NIPS 2017 adversarial example challenge [Kurakin et al., 2018], for example, highlights some of the progress made on developing classifiers that appear much stronger in practice than many of the ad-hoc techniques developed in previous years. Many of the approaches, though not formally verified in the strict sense during training, nonetheless have substantial theoretical justification for why they may perform well: Sinha et al. [2018] uses properties of statistical robustness to develop an approach that is not much more difficult to train and which empirically does achieve some measure of resistance to attacks; Madry et al. [2017] considers robustness to a first-order adversary, and shows that a randomized projected gradient descent procedure is optimal in this setting. Indeed, in some cases the classifiers trained via these methods can be verified to be adversarially robust using the verification techniques discussed below (though only for very small networks). Despite this progress, we believe it is also crucially important to consider defenses that *are* provably robust, to avoid any possible attack.

Second, our work in this paper relates closely to techniques for the formal verification of neural networks systems (indeed, our approach can be viewed as a convex procedure for verification, coupled with a method for training networks via the verified bounds). In this area, most past work focuses on using exact (combinatorial) solvers to verify the robustness properties of networks, either via Satisfiability Modulo Theories (SMT) solvers [Huang et al., 2017, Ehlers, 2017, Carlini and Wagner, 2017] or integer programming approaches [Lomuscio and Maganti, 2017, Tjeng and Tedrake, 2017, Cheng et al., 2017]. These methods have the benefit of being able to reason exactly about robustness, but at the cost of being combinatorial in complexity. This drawback has so far prevented these methods from effectively scaling to large models or being used within a training setting. There have also been a number of recent attempts to verify networks using non-combinatorial methods (and this current work fits broadly in this general area). For example, Gehr et al. [2018] develop a suite of verification methods based upon abstract interpretations (these can be broadly construed as relaxations of combinations of activations that are maintained as they pass through the network). Dvijotham et al. [2018] use an approach based upon analytically solving an optimization problem resulting from dual functions of the activations (which extends to activations beyond the ReLU). However, these methods apply to simple feedforward architectures without skip connections, and focus only on verification of existing networks.

Third, and most relevant to our current work, there are several approaches that go beyond provable verification, and also integrate the verification procedure into the training of the network itself. For example, Hein and Andriushchenko [2017] develop a formal bound for robustness to $\ell_2$ perturbations in two-layer networks, and train a surrogate of their bounds. Raghunathan et al. [2018] develop a semidefinite programming (SDP) relaxation of exact verification methods, and train a network by minimizing this bound via the dual SDP. And Wong and Kolter [2017] present a linear-programming (LP) based upper bound on the robust error or loss that can be suffered under norm-bounded

perturbation, then minimize this upper bound during training; the method is particularly efficient since they do not solve the LP directly, but instead show that it is possible to bound the LP optimal value and compute elementwise bounds on the activation functions based on a backward pass through the network. However, it is still the case that none of these approaches scale to realistically-sized networks; even the approach of [Wong and Kolter, 2017], which empirically has been scaled to the largest settings of all the above approaches, in the worst case scales *quadratically* in the number of hidden units in the network and dimensions in the input. Thus, all the approaches so far have been limited to relatively small networks and problems such as MNIST.

**Contributions**   This paper fits into this third category of integrating verification into training, and makes substantial progress towards scaling these methods to realistic settings. While we cannot yet reach e.g. ImageNet scales, even in this current work, we show that it *is* possible to overcome the main hurdles to scalability of past approaches. Specifically, we develop a provably robust training procedure, based upon the approach in [Wong and Kolter, 2017], but extending it in three key ways. The resulting method: 1) extends to general networks with skip connections, residual layers, and activations besides the ReLU; we do so by using a general formulation based on the Fenchel conjugate function of activations; 2) scales *linearly* in the dimensionality of the input and number of hidden units in the network, using techniques from nonlinear random projections, all while suffering minimal degradation in accuracy; and 3) further improves the quality of the bound with model cascades. We describe each of these contributions in the next section.

## 3   Scaling provably robust networks

### 3.1   Robust bounds for general networks via modular dual functions

This section presents an architecture for constructing provably robust bounds for general deep network architectures, using Fenchel duality. Importantly, we derive the dual of each network operation in a fully modular fashion, simplifying the problem of deriving robust bounds of a network to bounding the dual of individual functions. By building up a toolkit of dual operations, we can automatically construct the dual of any network architecture by iterating through the layers of the original network.

**The adversarial problem for general networks**   We consider a generalized $k$ "layer" neural network $f_\theta : \mathbb{R}^{|x|} \to \mathbb{R}^{|y|}$ given by the equations

$$z_i = \sum_{j=1}^{i-1} f_{ij}(z_j), \ \ \text{for } i = 2, \dots, k \tag{1}$$

where $z_1 = x$, $f_\theta(x) \equiv z_k$ (i.e., the output of the network) and $f_{ij} : \mathbb{R}^{|z_j|} \to \mathbb{R}^{|z_i|}$ is some function from layer $j$ to layer $i$. Importantly, this differs from prior work in two key ways. First, unlike the conjugate forms found in Wong and Kolter [2017], Dvijotham et al. [2018], we no longer assume that the network consists of linear operations followed by activation functions, and instead opt to work with an arbitrary sequence of $k$ functions. This simplifies the analysis of sequential non-linear activations commonly found in modern architectures, e.g. max pooling or a normalization strategy followed by a ReLU,[1] by analyzing each activation independently, whereas previous work would need to analyze the entire sequence as a single, joint activation. Second, we allow layers to depend not just on the previous layer, but also on all layers before it. This generalization applies to networks with any kind of skip connections, e.g. residual networks and dense networks, and greatly expands the set of possible architectures.

Let $\mathcal{B}(x) \subset \mathbb{R}^{|x|}$, represent some input constraint for the adversary. For this section we will focus on an arbitrary norm ball $\mathcal{B}(x) = \{x + \Delta : \|\Delta\| \le \epsilon\}$. This is the constraint set considered for norm-bounded adversarial perturbations, however other constraint sets can certainly be considered. Then, given an input example $x$, a known label $y^*$, and a target label $y^{\text{targ}}$, the problem of finding the most adversarial example within $\mathcal{B}$ (i.e., a so-called *targeted* adversarial attack) can be written as

$$\underset{z_k}{\text{minimize}} \ \ c^T z_k, \ \ \text{subject to} \ \ z_i = \sum_{j=1}^{i-1} f_{ij}(z_j), \ \text{for } i = 2, \dots, k, \ \ z_1 \in \mathcal{B}(x) \tag{2}$$

where $c = e_{y^\star} - e_{y^{\mathrm{targ}}}$.

**Dual networks via compositions of modular dual functions**    To bound the adversarial problem, we look to its dual optimization problem using the machinery of Fenchel conjugate functions [Fenchel, 1949], described in Definition 1.

**Definition 1.** *The conjugate of a function $f$ is another function $f^*$ defined by*

$$f^*(y) = \max_x x^T y - f(x) \tag{3}$$

Specifically, we can lift the constraint $z_{i+1} = \sum_{j=1}^i f_{ij}(z_j)$ from Equation 2 into the objective with an indicator function, and use conjugate functions to obtain a lower bound. For brevity, we will use the subscript notation $(\cdot)_{1:i} = ((\cdot)_1, \ldots, (\cdot)_i)$, e.g. $z_{1:i} = (z_1, \ldots, z_i)$. Due to the skip connections, the indicator functions are not independent, so we cannot directly conjugate each individual indicator function. We can, however, still form its dual using the conjugate of a different indicator function corresponding to the backwards direction, as shown in Lemma 1.

**Lemma 1.** *Let the indicator function for the $i$th constraint be*

$$\chi_i(z_{1:i}) = \left\{ \begin{array}{ll} 0 & \text{if } z_i = \sum_{j=1}^{i-1} f_{ij}(z_j) \\ \infty & \text{otherwise,} \end{array} \right. \tag{4}$$

*for $i = 2, \ldots, k$, and consider the joint indicator function $\sum_{i=2}^k \chi_i(z_{1:i})$. Then, the joint indicator is lower bounded by $\max_{\nu_{1:k}} \nu_k^T z_k - \nu_1^T z_1 - \sum_{i=1}^{k-1} \chi_i^*(-\nu_i, \nu_{i+1:k})$, where*

$$\chi_i^*(\nu_{i:k}) = \max_{z_i} \nu_i^T z_i + \sum_{j=i+1}^k \nu_j^T f_{ji}(z_i) \tag{5}$$

*for $i = 1, \ldots, k-1$. Note that $\chi_i^*(\nu_{i:k})$ is the exact conjugate of the indicator for the set $\{x_{i:k} : x_j = f_{ji}(x_i) \;\; \forall j > i\}$, which is different from the set indicated by $\chi_i$. However, when there are no skip connections (i.e. $z_i$ only depends on $z_{i-1}$), $\chi_i^*$ is exactly the conjugate of $\chi_i$.*

We defer the proof of Lemma 1 to Appendix A.1. With structured upper bounds on these conjugate functions, we can bound the original adversarial problem using the dual network described in Theorem 1. We can then optimize the bound using any standard deep learning toolkit using the same robust optimization procedure as in Wong and Kolter [2017] but using our bound instead. This amounts to minimizing the loss evaluated on our bound of possible network outputs under perturbations, as a drop in replacement for the traditional network output. For the adversarial setting, note that the $\ell_\infty$ perturbation results in a dual norm of $\ell_1$.

**Theorem 1.** *Let $g_{ij}$ and $h_i$ be any functions such that*

$$\chi_i^*(-\nu_i, \nu_{i+1:k}) \leq h_i(\nu_{i:k}) \;\; \text{subject to} \;\; \nu_i = \sum_{j=i+1}^k g_{ij}(\nu_j) \tag{6}$$

*for $i = 1, \ldots, k-1$. Then, the adversarial problem from Equation 2 is lower bounded by*

$$J(x, \nu_{1:k}) = -\nu_1^T x - \epsilon \|\nu_1\|_* - \sum_{i=1}^{k-1} h_i(\nu_{i:k}) \tag{7}$$

*where $\| \cdot \|_*$ is the dual norm, and $\nu_{1:k} = g(c)$ is the output of a $k$ layer neural network $g$ on input $c$, given by the equations*

$$\nu_k = -c, \;\; \nu_i = \sum_{j=i}^{k-1} g_{ij}(\nu_{j+1}), \;\; \text{for } i = 1, \ldots, k-1. \tag{8}$$

We denote the upper bound on the conjugate function from Equation 6 a *dual layer*, and defer the proof to Appendix A.2. To give a concrete example, we present two possible dual layers for linear operators and ReLU activations in Corollaries 1 and 2 (their derivations are in Appendix B), and we also depict an example dual residual block in Figure 1.

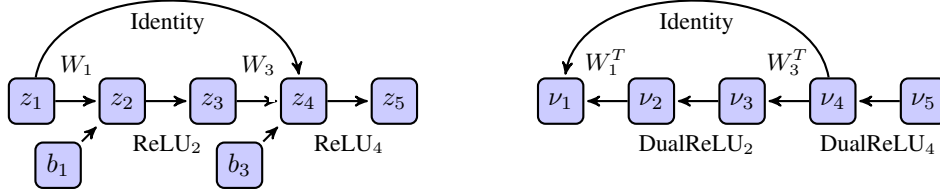

Figure 1: An example of the layers forming a typical residual block (left) and its dual (right), using the dual layers described in Corollaries 1 and 2. Note that the bias terms of the residual network go into the dual objective and are not part of the structure of the dual network, and the skip connections remain in the dual network but go in the opposite direction.

**Corollary 1.** *The dual layer for a linear operator* $\hat{z}_{i+1} = W_i z_i + b_i$ *is*

$$\chi_i^*(\nu_{i:k}) = \nu_{i+1}^T b_i \quad \text{subject to} \quad \nu_i = W_i^T \nu_{i+1}. \tag{9}$$

**Corollary 2.** *Suppose we have lower and upper bounds* $\ell_{ij}, u_{ij}$ *on the pre-activations. The dual layer for a ReLU activation* $\hat{z}_{i+1} = \max(z_i, 0)$ *is*

$$\chi_i^*(\nu_{i:k}) \leq -\sum_{j \in \mathcal{I}_i} \ell_{i,j}[\nu_{ij}]_+ \quad \text{subject to} \quad \nu_i = D_i \nu_{i+1}. \tag{10}$$

*where* $\mathcal{I}_i^-, \mathcal{I}_i^+, \mathcal{I}$ *denote the index sets where the bounds are negative, positive or spanning the origin respectively, and where* $D_i$ *is a diagonal matrix with entries*

$$(D_i)_{jj} = \begin{cases} 0 & j \in \mathcal{I}_i^- \\ 1 & j \in \mathcal{I}_i^+ \\ \frac{u_{i,j}}{u_{i,j} - \ell_{i,j}} & j \in \mathcal{I}_i \end{cases}. \tag{11}$$

We briefly note that these dual layers recover the original dual network described in Wong and Kolter [2017]. Furthermore, the dual linear operation is the exact conjugate and introduces no looseness to the bound, while the dual ReLU uses the same relaxation used in Ehlers [2017], Wong and Kolter [2017]. More generally, the strength of the bound from Theorem 1 relies entirely on the tightness of the individual dual layers to their respective conjugate functions in Equation 6. While any $g_{ij}, h_i$ can be chosen to upper bound the conjugate function, a tighter bound on the conjugate results in a tighter bound on the adversarial problem.

If the dual layers for all operations are linear, the bounds for all layers can be computed with a single forward pass through the dual network using a direct generalization of the form used in Wong and Kolter [2017] (due to their similarity, we defer the exact algorithm to Appendix F). By trading off tightness of the bound with computational efficiency by using linear dual layers, we can efficiently compute all bounds and construct the dual network one layer at a time. The end result is that we can automatically construct dual networks from dual layers in a fully modular fashion, completely independent of the overall network architecture (similar to how auto-differentiation tools proceed one function at a time to compute all parameter gradients using only the local gradient of each function). With a sufficiently comprehensive toolkit of dual layers, we can compute provable bounds on the adversarial problem for any network architecture.

For other dual layers, we point the reader to two resources. For the explicit form of dual layers for hardtanh, batch normalization, residual connections, we direct the reader to Appendix B. For analytical forms of conjugate functions of other activation functions such as tanh, sigmoid, and max pooling, we refer the reader to Dvijotham et al. [2018].

## 3.2 Efficient bound computation for $\ell_\infty$ perturbations via random projections

A limiting factor of the proposed algorithm and the work of Wong and Kolter [2017] is its computational complexity: for instance, to compute the bounds exactly for $\ell_\infty$ norm bounded perturbations in ReLU networks, it is computationally expensive to calculate $\|\nu_1\|_1$ and $\sum_{j \in \mathcal{I}_i} \ell_{ij}[\nu_{ij}]_+$. In contrast to other terms like $\nu_{i+1}^T b_i$ which require only sending a single bias vector through the dual network,

**Algorithm 1** Estimating $\|\nu_1\|_1$ and $\sum_{j \in \mathcal{I}} \ell_{ij}[\nu_{ij}]_+$

---

**input:** Linear dual network operations $g_{ij}$, projection dimension $r$, lower bounds $\ell_{ij}$, $d_{ij}$ from Equation 13, layer-wise sizes $|z_i|$

$R_1^{(1)} := \text{Cauchy}(r, |z_1|)$ // *initialize random matrix for $\ell_1$ term*

**for** $i = 2, \ldots, k$ **do**

    // *pass each term forward through the network*

    **for** $j = 1, \ldots, i-1$ **do**

        $R_j^{(i)}, S_j^{(i)} := \sum_{k=1}^{i-1} g_{ki}^T(R_i^{(k)}), \sum_{k=1}^{i-1} g_{ki}^T(S_i^{(k)})$

    **end for**

    $R_i^{(i)}, S_i^{(i)} := \text{diag}(d_i)\text{Cauchy}(|z_i|, r), d_i$ // *initialize terms for layer $i$*

**end for**

**output:** $\text{median}(|R_1^{(k)}|), 0.5\left(-\text{median}(|R_2^{(k)}|) + S_2^{(k)}\right), \ldots, 0.5\left(-\text{median}(|R_k^{(k)}|) + S_k^{(k)}\right)$

---

the matrices $\nu_1$ and $\nu_{i,\mathcal{I}_i}$ must be explicitly formed by sending an example through the dual network for each input dimension and for each $j \in \mathcal{I}_i$, which renders the entire computation *quadratic* in the number of hidden units. To scale the method for larger, ReLU networks with $\ell_\infty$ perturbations, we look to random Cauchy projections. Note that for an $\ell_2$ norm bounded adversarial perturbation, the dual norm is also an $\ell_2$ norm, so we can use traditional random projections [Vempala, 2005]. Experiments for the $\ell_2$ norm are explored further in Appendix H. However, for the remainder of this section we focus on the $\ell_1$ case arising from $\ell_\infty$ perturbations.

**Estimating with Cauchy random projections** From the work of Li et al. [2007], we can use the sample median estimator with Cauchy random projections to directly estimate $\|\nu_1\|_1$ for linear dual networks, and use a variation to estimate $\sum_{j \in \mathcal{I}} \ell_{ij}[\nu_{ij}]_+$, as shown in Theorem 2 (the proof is in Appendix D.1).

**Theorem 2.** *. Let $\nu_{1:k}$ be the dual network from Equation 1 with linear dual layers and let $r > 0$ be the projection dimension. Then, we can estimate*

$$\|\nu_1\|_1 \approx \text{median}(|\nu_1^T R|) \tag{12}$$

*where $R$ is a $|z_1| \times r$ standard Cauchy random matrix and the median is taken over the second axis. Furthermore, we can estimate*

$$\sum_{j \in \mathcal{I}} \ell_{ij}[\nu_{ij}]_+ \approx \frac{1}{2}\left(-\text{median}(|\nu_i^T \text{diag}(d_i)R|) + \nu_i^T d_i\right), \quad d_{i,j} = \begin{cases} \frac{u_{i,j}}{u_{i,j} - \ell_{i,j}} & j \notin \mathcal{I}_i \\ 0 & j \in \mathcal{I}_i \end{cases} \tag{13}$$

*where $R$ is a $|z_i| \times r$ standard Cauchy random matrix, and the median is taken over the second axis.*

This estimate has two main advantages: first, it is simple to compute, as evaluating $\nu_1^T R$ involves passing the random matrix forward through the dual network (similarly, the other term requires passing a modified random matrix through the dual network; the exact algorithm is detailed in 1). Second, it is memory efficient in the backward pass, as the gradient need only propagate through the median entries.

These random projections reduce the computational complexity of computing these terms to piping $r$ random Cauchy vectors (and an additional vector) through the network. Crucially, the complexity is no longer a quadratic function of the network size: if we fix the projection dimension to some constant $r$, then the computational complexity is now linear with the input dimension and $\mathcal{I}_i$. Since previous work was either quadratic or combinatorially expensive to compute, estimating the bound with random projections is the fastest and most scalable approach towards training robust networks that we are aware of. At test time, the bound can be computed exactly, as the gradients no longer need to be stored. However, if desired, it is possible to use a different estimator (specifically, the geometric estimator) for the $\ell_\infty$ norm to calculate high probability bounds on the adversarial problem, which is discussed in Appendix E.1.

### 3.3 Bias reduction with cascading ensembles

A final major challenge of training models to minimize a robust bound on the adversarial loss, is that the robustness penalty acts as a regularization. For example, in a two-layer ReLU network, the robust

Table 1: Number of hidden units, parameters, and time per epoch for various architectures.

| Model | Dataset | # hidden units | # parameters | Time (s) / epoch |
|---|---|---|---|---|
| Small | MNIST | 4804 | 166406 | 74 |
|  | CIFAR | 6244 | 214918 | 48 |
| Large | MNIST | 28064 | 1974762 | 667 |
|  | CIFAR | 62464 | 2466858 | 466 |
| Resnet | MNIST | 82536 | 3254562 | 2174 |
|  | CIFAR | 107496 | 4214850 | 1685 |

Table 2: Results on MNIST, and CIFAR10 with small networks, large networks, residual networks, and cascaded variants.

| Dataset | Model | Epsilon | Single model error Robust | Single model error Standard | Cascade error Robust | Cascade error Standard |
|---|---|---|---|---|---|---|
| MNIST | Small, Exact | 0.1 | 4.48% | 1.26% | - | - |
| MNIST | Small | 0.1 | 4.99% | 1.37% | **3.13%** | **3.13%** |
| MNIST | Large | 0.1 | **3.67%** | **1.08%** | 3.42% | 3.18% |
| MNIST | Small | 0.3 | **43.10%** | 14.87% | **33.64%** | **33.64%** |
| MNIST | Large | 0.3 | 45.66% | **12.61%** | 41.62% | 35.24% |
| CIFAR10 | Small | 2/255 | 52.75% | 38.91% | 39.35% | 39.35% |
| CIFAR10 | Large | 2/255 | 46.59% | **31.28%** | 38.84% | 36.08% |
| CIFAR10 | Resnet | 2/255 | **46.11%** | 31.72% | **36.41%** | **35.93%** |
| CIFAR10 | Small | 8/255 | 79.25% | 72.24% | 71.71% | 71.71% |
| CIFAR10 | Large | 8/255 | 83.43% | 80.56 | 79.24% | 79.14% |
| CIFAR10 | Resnet | 8/255 | **78.22%** | **71.33%** | **70.95%** | **70.77%** |

loss penalizes $\epsilon\|\nu_1\|_1 = \epsilon\|W_1 D_1 W_2\|_1$, which effectively acts as a regularizer on the network with weight $\epsilon$. Because of this, the resulting networks (even those with large representational capacity), are typically overregularized to the point that many filters/weights become identically zero (i.e., the network capacity is not used).

To address this point, we advocate for using a robust *cascade* of networks: that is, we train a sequence of robust classifiers, where later elements of the cascade are trained (and evaluated) *only on those examples that the previous elements of the cascade cannot certify* (i.e., those examples that lie within $\epsilon$ of the decision boundary). This procedure is formally described in the Appendix in Algorithm 2.

## 4 Experiments

**Dataset and Architectures** We evaluate the techniques in this paper on two main datasets: MNIST digit classification [LeCun et al., 1998] and CIFAR10 image classification [Krizhevsky, 2009].[2] We test on a variety of deep and wide convolutional architectures, with and without residual connections. All code for these experiments is available at `https://github.com/locuslab/convex_adversarial/`. The small network is the same as that used in [Wong and Kolter, 2017], with two convolutional layers of 16 and 32 filters and a fully connected layer of 100 units. The large network is a scaled up version of it, with four convolutional layers with 32, 32, 64, and 64 filters, and two fully connected layers of 512 units. The residual networks use the same structure used by [Zagoruyko and Komodakis, 2016] with 4 residual blocks with 16, 16, 32, and 64 filters. We highlight a subset of the results in Table 2, and briefly describe a few key observations below. We leave more extensive experiments and details regarding the experimental setup in Appendix G, including additional experiments on $\ell_2$ perturbations. All results except where otherwise noted use random projection of 50 dimensions.

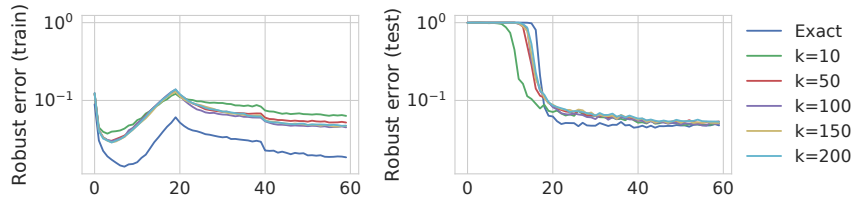

Figure 2: Training and testing robust error curves over epochs on the MNIST dataset using $k$ projection dimensions. The $\epsilon$ value for training is scheduled from 0.01 to 0.1 over the first 20 epochs. The projections force the model to generalize over higher variance, reducing the generalization gap.

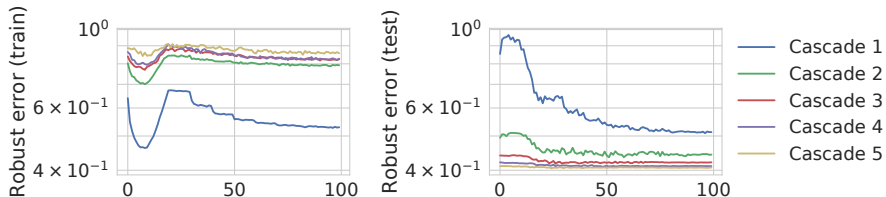

Figure 3: Robust error curves as we add models to the cascade for the CIFAR10 dataset on a small model. The $\epsilon$ value for training is scheduled to reach 2/255 after 20 epochs. The training curves are for each individual model, and the testing curves are for the whole cascade up to the stage.

**Summary of results** For the different data sets and models, the final robust and nominal test errors are given in Table 2. We emphasize that in all cases we report the *robust test error*, that is, our *upper bound* on the possible test set error that the classifier can suffer under *any* norm-bounded attack (thus, considering different empirical attacks is orthogonal to our main presentation and not something that we include, as we are focused on verified performance). As we are focusing on the particular random projections discussed above, all experiments consider attacks with bounded $\ell_\infty$ norm, plus the ReLU networks highlighted above. On MNIST, the (non-cascaded) large model reaches a final robust error of 3.7% for $\epsilon = 0.1$, and the best cascade reaches 3.1% error. This contrasts with the best previous bound of 5.8% robust error for this epsilon, from [Wong and Kolter, 2017]. On CIFAR10, the ResNet model achieves 46.1% robust error for $\epsilon = 2/255$, and the cascade lowers this to 36.4% error. In contrast, the previous best *verified* robust error for this $\epsilon$, from [Dvijotham et al., 2018], was 80%. While the robust error is naturally substantially higher for $\epsilon = 8/255$ (the amount typically considered in empirical works), we are still able to achieve 71% provable robust error; for comparison, the best *empirical* robust performance against current attacks is 53% error at $\epsilon = 8/255$ Madry et al. [2017], and most heuristic defenses have been broken to beyond this error Athalye et al. [2018].

**Number of random projections** In the MNIST dataset (the only data set where it is trivial to run exact training without projection), we have evaluated our approach using different projection dimensions as well as exact training (i.e., without random projections). We note that using substantially lower projection dimension does not have a significant impact on the test error. This fact is highlighted in Figure 2. Using the same convolutional architecture used by Wong and Kolter [2017], which previously required gigabytes of memory and took hours to train, it is sufficient to use only 10 random projections to achieve comparable test error performance to training with the exact bound. Each training epoch with 10 random projections takes less than a minute on a single GeForce GTX 1080 Ti graphics card, while using less than 700MB of memory, achieving significant speedup and memory reduction over Wong and Kolter [2017]. The estimation quality and the corresponding speedups obtained are explored in more detail in Appendix E.6.

**Cascades** Finally, we consider the performance of the cascaded versus non-cascaded models. In all cases, cascading the models is able to improve the robust error performance, sometimes substantially, for instance decreasing the robust error on CIFAR10 from 46.1% to 36.4% for $\epsilon = 2/255$. However, this comes at a cost as well: the *nominal* error *increases* throughout the cascade (this is to be expected, since the cascade essentially tries to force the robust and nominal errors to match). Thus, there is

substantial value to both improving the single-model networks *and* integrating cascades into the prediction.

## 5 Conclusion

In this paper, we have presented a general methodology for deriving dual networks from compositions of dual layers based on the methodology of conjugate functions to train classifiers that are provably robust to adversarial attacks. Importantly, the methodology is linearly scalable for ReLU based networks against $\ell_\infty$ norm bounded attacks, making it possible to train large scale, provably robust networks that were previously out of reach, and the obtained bounds can be improved further with model cascades. While this marks a significant step forward in scalable defenses for deep networks, there are several directions for improvement. One particularly important direction is better architecture development: a wide range of functions and activations not found in traditional deep residual networks may have better robustness properties or more efficient dual layers that also allow for scalable training. But perhaps even more importantly, we also need to consider the nature of adversarial perturbations beyond just norm-bounded attacks. Better characterizing the space of perturbations that a network "should" be resilient to represents one of the major challenges going forward for adversarial machine learning.

## Footnotes

[1]Batch normalization, since it depends on entire minibatches, is formally not covered by the approach, but it can be approximated by considering the scaling and shifting to be generic parameters, as is done at test time.

[2]We fully realize the irony of a paper with "scaling" in the title that currently maxes out on CIFAR10 experiments. But we emphasize that when it comes to certifiably robust networks, the networks we consider here, as we illustrate below in Table 1, are more than an order of magnitude larger than any that have been considered previously in the literature. Thus, our emphasis is really on the potential scaling properties of these approaches rather than large-scale experiments on e.g. ImageNet sized data sets.

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
