[Supplementary Material]

# A  Conjugates and lower bounds with duality

## A.1  Conjugates of the joint indicator function

Here, we derive a lower bound on $\sum_{i=2}^{k} \chi_i(z_{1:i})$. It is mathematically convenient to introduce addition variables $\hat{z}_{1:k}$ such that $\hat{z}_i = z_i$ for all $i = 1, \ldots, k$, and rephrase it as the equivalent constrained optimization problem.

$$\min_{z_{1:k-1}, \hat{z}_{2:k}} 0$$

$$\text{subject to } \hat{z}_i = \sum_{j=1}^{i-1} f_{ij}(z_j) \quad \text{for} \quad i = 2, \ldots, k \tag{14}$$

$$z_i = \hat{z}_i \quad \text{for } i = 1, \ldots, k$$

Note that we do not optimize over $\hat{z}_1$ and $z_k$ yet, to allow for future terms on the inputs and outputs of the network, so this is analyzing just the network structure. We introduce Lagrangian variables $\nu_{1:k}, \hat{\nu}_{2:k}$ to get the following Lagrangian:

$$L(z_{1:k}, \hat{z}_{1:k}, \nu_{1:k}, \hat{\nu}_{2:k}) = \sum_{i=2}^{k} \hat{\nu}_i^T \left( \hat{z}_i - \sum_{j=1}^{i-1} f_{ij}(z_j) \right) + \sum_{i=1}^{k} \nu_i^T (z_i - \hat{z}_i) \tag{15}$$

Grouping up terms by $z_i, \hat{z}_i$ and rearranging the double sum results in the following expression:

$$L(z_{1:k}, \hat{z}_{1:k}, \nu_{1:k}, \hat{\nu}_{2:k}) = -\nu_1^T \hat{z}_1 + \sum_{i=2}^{k} (\hat{\nu}_i - \nu_i)^T \hat{z}_i + \sum_{i=1}^{k} \left( \nu_i^T z_i - \sum_{j=i+1}^{k} \hat{\nu}_j^T f_{ji}(z_i) \right) \tag{16}$$

From the KKT stationarity conditions for the derivative with respect to $\hat{z}_i$, we know that $\hat{\nu}_i = \nu_i$. Also note that in the summand, the last term for $i = k$ has no double summand, so we move it out for clarity.

$$L(z_{1:k}, \nu_{1:k}) = -\nu_1^T \hat{z}_1 + \nu_k^T z_k + \sum_{i=1}^{k-1} \left( \nu_i^T z_i - \sum_{j=i+1}^{k} \nu_j^T f_{ji}(z_i) \right) \tag{17}$$

Finally, we minimize over $z_i$ for $i = 2, \ldots, k-1$ to get the conjugate form for the lower bound via weak duality.

$$L(z_{1:k}, \nu_{1:k}) \geq -\nu_1^T \hat{z}_1 + \nu_k^T z_k + \sum_{i=1}^{k-1} \min_{z_i} \left( \nu_i^T z_i - \sum_{j=i+1}^{k} \nu_j^T f_{ji}(z_i) \right)$$

$$= -\nu_1^T \hat{z}_1 + \nu_k^T z_k - \sum_{i=1}^{k-1} \max_{z_i} \left( -\nu_i^T z_i + \sum_{j=i+1}^{k} \nu_j^T f_{ji}(z_i) \right) \tag{18}$$

$$= -\nu_1^T z_1 + \nu_k^T z_k - \sum_{i=1}^{k-1} \chi_i^*(-\nu_i, \nu_{i+1:k})$$

## A.2  Proof of Theorem 1

First, we rewrite the primal problem by bringing the function and input constraints into the objective with indicator functions $I$. We can then apply Lemma 1 to get the following lower bound on the adversarial problem:

$$\underset{\nu_{1:k}}{\text{maximize}} \, \underset{z_1, z_k}{\text{minimize}} \, (c^T + \nu_k)^T z_k + I_{\mathcal{B}(x)}(z_1) - \nu_1^T z_1 - \sum_{i=1}^{k-1} \chi_i^*(-\nu_i, \nu_{i+1:k}) \tag{19}$$

Minimizing over $z_1$ and $z_k$, note that

$$\min_{\hat{z}_k} (c + \nu_k)^T \hat{z}_k = -I(\nu_k = -c)$$

$$\min_{\hat{z}_1} I_{\mathcal{B}(x)}(z_1) - \nu_1^T z_1 = -I_{\mathcal{B}(x)}^*(\nu_1) \tag{20}$$

Note that if $\mathcal{B}(x) = \{x + \Delta : \|\Delta\| \leq \epsilon\}$ for some norm, then $I^*_{\mathcal{B}(x)}(\nu_1) = \nu_1^T x + \epsilon \|\nu_1\|_*$ where $\|\cdot\|$ is the dual norm, but any sort of input constraint can be used so long as its conjugate can be bounded. Finally, the last term can be bounded with the dual layer:

$$\min_{z_i} \nu_i^T z_i - \sum_{j=i+1}^{k} \nu_j^T f_{ji}(z_i) = -\chi_i^*(-\nu_i, \nu_{i+1:k}) \geq -h_i(\nu_{i:k}) \quad \text{subject to} \quad \nu_i = \sum_{j=i+1}^{k} g_{ij}(\nu_j)$$
(21)

Combining these all together, we get that the adversarial problem from Equation 2 is lower bounded by

$$\begin{aligned} \underset{\nu}{\text{maximize}} \quad & -\nu_1^T x - \epsilon \|\nu_1\|_* - \sum_{i=1}^{k-1} h_i(\nu_{i:k}) \\ \text{subject to} \quad & \nu_k = -c \\ & \nu_i = \sum_{j=i+1}^{k} g_{ij}(\nu_j) \end{aligned}$$
(22)

## B   Dual layers

In this section, we derive the dual layers for standard building blocks of deep learning.

### B.1   Linear operators

Suppose $f_i(z_i) = W_i z_i + b_i$ for some linear operator $W_i$ and bias terms $b_i$. Then,

$$\begin{aligned} \chi_i^*(-\nu_i, \nu_{i+1}) &= \max_{z_i} -z_i^T \nu_i + (W_i z_i + b_i)^T \nu_{i+1} \\ &= \max_{z_i} z_i^T (W_i^T \nu_{i+1} - \nu_i) + b_i^T \nu_{i+1} \\ &= \max_{z_i} I(\nu_i = W_i^T \nu_{i+1}) + b_i^T \nu_{i+1} \\ &= b_i^T \nu_{i+1} \quad \text{subject to} \quad \nu_i = W_i^T \nu_{i+1} \end{aligned}$$
(23)

### B.2   Residual linear connections

Suppose $f_i(z_i, z_j) = W_i z_i + z_j + b_i$ and $z_{j+1} = W_j z_j + b_j$ for some $j < i - 1$ for linear operators $W_i, W_j$ and bias term $b_i, b_j$. Then,

$$\begin{aligned} \chi_i^*(-\nu_i, \nu_{i+1}) &= \max_{z_i} -z_i^T \nu_i + (W_i z_i + b_i)^T \nu_{i+1} \\ &= b_i^T \nu_{i+1} \quad \text{subject to} \quad \nu_i = W_i^T \nu_{i+1} \end{aligned}$$
(24)

and

$$\begin{aligned} \chi_i^*(-\nu_j, \nu_{j+1}) &= \max_{z_j} -z_j^T \nu_j + z_j^T \nu_i + (W_j z_j + b_j)^T \nu_{j+1} \\ &= b_j^T \nu_j \quad \text{subject to} \quad \nu_j = W_j^T \nu_{j+1} + \nu_i \end{aligned}$$
(25)

### B.3   ReLU activations

The proof here is the same as that presented in Appendix A3 of Wong and Kolter [2017], however we reproduce a simplified version here for the reader. The conjugate function for the ReLU activation is the following:

$$\chi^*(-\nu_i, \nu_{i+1}) = \max_{z_i} -z_i^T \nu_i + \max(z_i, 0)\nu_{i+1}$$
(26)

Suppose we have lower and upper bounds $\ell_i, u_i$ on the input $z_i$. If $u_i \leq 0$, then $\max(z_i, 0) = 0$, and so

$$\chi^*(-\nu_i, \nu_{i+1}) = \max_{z_i} -z_i^T \nu_i = 0 \quad \text{subject to} \quad \nu_i = 0$$
(27)

Otherwise, if $\ell_i \geq 0$, then $\max(z_i, 0) = z_i$ and we have

$$\chi^*(-\nu_i, \nu_{i+1}) = \max_{z_i} -z_i^T \nu_i + z_i^T \nu_{i+1} = 0 \quad \text{subject to} \quad \nu_i = \nu_{i+1} \tag{28}$$

Lastly, suppose $\ell_i < 0 < u_i$. Then, we can upper bound the conjugate by taking the maximum over a convex outer bound of the ReLU, namely $\mathcal{S}_i = \{(z_i, z_{i+1}) : z_{i+1} \geq 0, z_{i+1} \geq z_i, -u_i \odot z_i + (u_i - \ell_i) \odot z_{i+1} \leq -u_i \odot \ell_i\}$, where $\odot$ denotes element-wise multiplication:

$$\chi^*(-\nu_i, \nu_{i+1}) \leq \max_{\mathcal{S}_i} -z_i^T \nu_i + z_{i+1}^T \nu_{i+1} \tag{29}$$

The maximum must occur either at the origin $(0, 0)$ or along the line $-u_{ij} z_{ij} + (u_{ij} - \ell_{ij}) z_{i+1,j} = -u_{ij} \ell_{ij}$, so we can upper bound it again with

$$
\begin{aligned}
\chi^*(-\nu_{ij}, \nu_{i+1,j}) &\leq \max_{z_{ij}} \left[ -z_{ij}\nu_{ij} + \left( \frac{u_{ij}}{u_{ij} - \ell_{ij}} z_{ij} - \frac{u_{ij}\ell_{ij}}{u_{ij} - \ell_{ij}} \right) \nu_{i+1,j} \right]_+ \\
&= \max_{z_{ij}} \left[ \left( \frac{u_{ij}}{u_{ij} - \ell_{ij}} \nu_{i+1,j} - \nu_{ij} \right) z_{ij} - \frac{u_{ij}\ell_{ij}}{u_{ij} - \ell_{ij}} \nu_{i+1,j} \right]_+ \\
&= \left[ -\frac{u_{ij}\ell_{ij}}{u_{ij} - \ell_{ij}} \nu_{i+1,j} \right]_+ \quad \text{subject to} \quad \nu_{ij} = \frac{u_{ij}}{u_{ij} - \ell_{ij}} \nu_{i+1,j} \\
&= -\ell_{ij} [\nu_{ij}]_+ \quad \text{subject to} \quad \nu_{ij} = \frac{u_{ij}}{u_{ij} - \ell_{ij}} \nu_{i+1,j}
\end{aligned}
\tag{30}
$$

Let $\mathcal{I}_i^-, \mathcal{I}_i^+, \mathcal{I}$ and $D_i$ be as defined in the corollary. Combining these three cases together, we get the final upper bound:

$$\chi_i^*(-\nu_i, \nu_{i+1:k}) \leq -\sum_{j \in \mathcal{I}_i} \ell_{i,j} [\nu_{i,j}]_+ \quad \text{subject to} \quad \nu_i = D_i \nu_{i+1} \tag{31}$$

## B.4 Hardtanh

Here, we derive a dual layer for the hardtanh activation function. The hard tanh activation function is given by

$$
\text{hardtanh}(x) = \begin{cases} -1 & \text{for} \quad x < -1 \\ x & \text{for} \quad -1 \leq x \leq 1 \\ 1 & \text{for} \quad x > 1 \end{cases}
\tag{32}
$$

Since this is an activation function (and has no skip connections), we only need to bound the following:

$$\chi^*(-\nu_i, \nu_{i+1}) = \max_{z_i} -z_i^T \nu_i + \text{hardtanh}(z_i)^T \nu_{i+1} \tag{33}$$

Given lower and upper bounds $\ell$ and $u$, we can use a similar convex relaxation as that used for ReLU and decompose this problem element-wise (we will now assume all terms are scalars for notational simplicity), so we have

$$\chi^*(\nu_i, \nu_{i+1}) \leq \max_{z_i, z_{i+1} \in \mathcal{S}} -z_i \nu_i + z_{i+1} \nu_{i+1} \tag{34}$$

where $\mathcal{S}$ is the convex relaxation. The exact form of the relaxation depends on the values of $\ell$ and $u$, and we proceed to derive the dual layer for each case. We depict the relaxation where $u > 1$ and $\ell < -1$ in Figure 4, and note that the remaining cases are either triangular relaxations similar to the ReLU case or exact linear regions.

### B.4.1 $u > 1, \ell < -1$

If $u > 1$ and $\ell < -1$, we can use the relaxation given in Figure 4. The upper bound goes through the points $(\ell, -1)$ and $(1, 1)$ while the lower bound goes through the points $(-1, -1)$ and $(u, 1)$. The slope of the first one is $\frac{2}{1-\ell}$ and the slope of the second one is $\frac{2}{u+1}$, so we have either

$$z_{i+1} = \frac{2}{1-\ell}(z_i - 1) + 1, \quad z_{i+1} = \frac{2}{u+1}(z_i + 1) - 1 \tag{35}$$

Figure 4: Convex relaxation of hardtanh given lower and upper bounds $\ell$ and $u$.

Taking the maximum over these two cases, we have our upper bound of the conjugate is

$$\chi^*(\nu_i, \nu_{i+1}) \leq \max\left(-z_i\nu_i + \left(\frac{2}{1-\ell}(z_i - 1) + 1\right)\nu_{i+1}, -z_i\nu_i + \left(\frac{2}{u+1}(z_i+1) - 1\right)\nu_{i+1}\right)$$
(36)

Simplifying we get

$$\chi^*(\nu_i, \nu_{i+1}) \leq \max\left(z_i\left(-\nu_i + \frac{2}{1-\ell}\nu_{i+1}\right) + \left(1 - \frac{2}{1-\ell}\right)\nu_{i+1},\right.$$
$$\left. z_i\left(-\nu_i + \frac{2}{u+1}\nu_{i+1}\right) + \left(\frac{2}{u+1} - 1\right)\nu_{i+1}\right)$$
(37)

So each case becomes

$$\chi^*(\nu_i, \nu_{i+1}) \leq \max\left(\left(1 - \frac{2}{1-\ell}\right)\nu_{i+1} \quad \text{subject to} \quad \nu_i = \frac{2}{1-\ell}\nu_{i+1},\right.$$
$$\left. \left(\frac{2}{u+1} - 1\right)\nu_{i+1} \quad \text{subject to} \quad \nu_i = \frac{2}{u+1}\nu_{i+1}\right)$$
(38)

As a special case, note that when $u = -\ell$, we have

$$\chi^*(\nu_i, \nu_{i+1}) \leq \left|\left(1 - \frac{2}{1+u}\right)\nu_{i+1}\right| \quad \text{subject to} \quad \nu_i = \frac{2}{1+u}\nu_{i+1}$$
(39)

This dual layer is linear, and so we can continue to use random projections for efficient bound estimation.

**B.4.2** $u \leq -1$

Then, $\mathcal{S} = \{z_{i+1} = -1\}$ and so

$$\chi^*(\nu_i, \nu_{i+1}) = \max_{z_i} -z_i\nu_i - \nu_{i+1} = -\nu_{i+1} \quad \text{subject to} \quad \nu_i = 0$$
(40)

**B.4.3** $\ell \geq 1$

Then, $\mathcal{S} = \{z_{i+1} = 1\}$ and so

$$\chi^*(\nu_i, \nu_{i+1}) = \max_{z_i} -z_i\nu_i + \nu_{i+1} = \nu_{i+1} \quad \text{subject to} \quad \nu_i = 0$$
(41)

**B.4.4** $\ell \geq -1, u \leq 1$

Then, $\mathcal{S} = \{z_{i+1} = z_i\}$ and so

$$\chi^*(\nu_i, \nu_{i+1}) = \max_{z_i} -z_i\nu_i + z_i\nu_{i+1} = 0 \quad \text{subject to} \quad \nu_i = \nu_{i+1}$$
(42)

**B.4.5** $\ell \le -1, -1 \le u \le 1$

Here, our relaxation consists of the triangle above the hardtanh function. Then, the maximum occurs either on the line $z_{i+1} = \frac{1+u}{u-\ell}(z_i - \ell) - 1$ or at $(-1,-1)$. This line is equivalent to $z_{i+1} = \frac{1+u}{u-\ell}z_i - \left(\frac{1+u}{u-\ell}\ell + 1\right)$, and the point $(-1,-1)$ has objective value $\nu_i - \nu_{i+1}$, so we get

$$\chi^*(\nu_i, \nu_{i+1}) \le \max_{z_i} -z_i\nu_i + \frac{1+u}{u-\ell}z_i\nu_{i+1} - \left(\frac{1+u}{u-\ell}\ell + 1\right)\nu_{i+1} \tag{43}$$

$$\chi^*(\nu_i, \nu_{i+1}) \le \max\left(-\left(\frac{1+u}{u-\ell}\ell + 1\right)\nu_{i+1}, \nu_i - \nu_{i+1}\right) \quad \text{subject to} \quad \nu_i = \frac{1+u}{u-\ell}\nu_{i+1} \tag{44}$$

**B.4.6** $-1 \le \ell \le 1, 1 \le u$

Here, our relaxation consists of the triangle below the hardtanh function. Then, the maximum occurs either on the line $z_{i+1} = \frac{1-\ell}{u-\ell}(z_i - \ell) + \ell$ or at $(1,1)$. This line is equivalent to $z_{i+1} = \frac{1-\ell}{u-\ell}z_i - \left(\frac{1-\ell}{u-\ell}\ell - \ell\right)$, and at the point $(1,1)$ has objective value $-\nu_i + \nu_{i+1}$, so we get

$$\chi^*(\nu_i, \nu_{i+1}) \le \max_{z_i} -z_i\nu_i + \frac{1-\ell}{u-\ell}z_i\nu_{i+1} - \left(\frac{1-\ell}{u-\ell}\ell - \ell\right)\nu_{i+1} \tag{45}$$

$$\chi^*(\nu_i, \nu_{i+1}) \le \max\left(-\left(\frac{1-\ell}{u-\ell}\ell - \ell\right)\nu_{i+1}, -\nu_i + \nu_{i+1}\right) \quad \text{subject to} \quad \nu_i = \frac{1-\ell}{u-\ell}\nu_{i+1} \tag{46}$$

**B.5 Batch normalization**

As mentioned before, we only consider the case of batch normalization with a fixed mean and variance. This is true during test time, and at training time we can use the batch statistics as a heuristic. Let $\mu_i, \sigma_i$ be the fixed mean and variance statistics, so batch normalization has the following form:

$$BN(z_i) = \gamma \frac{x_i - \mu_i}{\sqrt{\sigma_i^2 + \epsilon}} + \beta \tag{47}$$

where $\gamma, \beta$ are the batch normalization parameters. Then,

$$z_i = \gamma \frac{\hat{z}_i - \mu}{\sqrt{\sigma^2 + \epsilon}} + \beta = D_i z_i + d_i \tag{48}$$

where $D_{i+1} = \text{diag}\left(\frac{\gamma}{\sqrt{\sigma^2 + \epsilon}}\right)$ and $d_{i+1} = \beta - \frac{\mu}{\sqrt{\sigma^2 + \epsilon}}$. and so we can simply plug this into the linear case to get

$$\chi_i^*(-\nu_i, \nu_{i+1:k}) = d_i^T \nu_{i+1} \quad \text{subject to} \quad \nu_i = D_i \nu_{i+1} \tag{49}$$

Note however, that batch normalization has the effect of shifting the activations to be centered more around the origin, which is exactly the case in which the robust bound becomes looser. In practice, we find that while including batch normalization may improve convergence, it reduces the quality of the bound.

# C Cascade construction

The full algorithm for constructing cascades as we describe in the main text is shown in Algorithm 2. To illustrate the use of the cascade, Figure 5 shows a two stage cascade on a few data points in two dimensional space. The boxes denote the adversarial ball around each example, and if the decision boundary is outside of the box, the example is certified.

**Algorithm 2** Training robust cascade of $k$ networks and making predictions

---

**input:** Initialized networks $f_1, \ldots, f_k$, training examples $X, y$, robust training procedure denoted RobustTrain, test example $x^*$

**for** $i = 1, \ldots, k$ **do**

　$f_i := \text{RobustTrain}(f_i, X, y)$ *// Train network*

　*// remove certified examples from dataset*

　$X, y := \{x_i, y_i : J(x, g(e_{f(x_i)} - e_{y^{targ}})) > 0, \ \forall y^{targ} \neq f(x_i)\}$

**end for**

**for** $i = 1, \ldots, k$ **do**

　**if** $J(x, g(e_{f_i(x^*)} - e_{y^{targ}})) < 0 \ \forall y^{targ} \neq f_i(x^*)$ **then**

　　**output:** $f_i(x^*)$ *// return label if certified*

　**end if**

**end for**

**output:** no certificate

---

Figure 5: An example of a two stage cascade. The first model on the left can only robustly classify three of the datapoints. After removing the certified examples, the remaining examples can now easily be robustly classified by a second stage classifier.

## D　Estimation using Cauchy random projections

### D.1　Proof of Theorem 2

**Estimating** $\|\hat{\nu}_1\|_{1,:}$　Recall the form of $\hat{\nu}_1$,

$$\hat{\nu}_1 = I W_1^T D_2 W_2^T \ldots D_n W_n^T = g(I)$$

where we include the identity term to make explicit the fact that we compute this by passing an identity matrix through the network $g$. Estimating this term is straightforward: we simply pass in a Cauchy random matrix $R$, and take the median absolute value:

$$\|\hat{\nu}_1\|_{1,:} \approx \text{median}(|R W_1^T D_2 W_2^T \ldots D_n W_n^T|) = \text{median}(|g(R)|)$$

where the median is taken over the minibatch axis.

**Estimating** $\sum_i [\nu_{i,:}]_+$　Recall the form of $\nu = \nu_j$ for some layer $j$,

$$\nu_j = I D_j W_j^T \ldots D_n W_n^T = g_j(I)$$

Note that for a vector $x$,

$$\sum_i [x]_+ = \frac{\|x\|_1 + 1^T x}{2}$$

So we can reuse the $\ell_1$ approximation from before to get

$$\sum_i [\nu_{i,:}]_+ = \frac{\|\nu\|_{1,:} + 1^T \nu}{2} \approx \frac{|\text{median}(g_j(R)) + g_j(1^T)|}{2}$$

which involves using the same median estimator and also passing in a single example of ones through the network.

**Estimating** $\sum_{i \in \mathcal{I}} \ell_i[\nu_{i,:}]_+$  The previous equation, while simple, is not exactly the term in the objective; there is an addition $\ell_1$ factor for each row, and we only add rows in the $\mathcal{I}$ set. However, we can deal with this by simply passing in a modified input to the network, as we will see shortly:

$$
\begin{aligned}
\sum_{i \in \mathcal{I}} \ell_i[\nu_{i,:}]_+ &= \sum_{i \in \mathcal{I}} \ell_i \frac{|\nu_{i,:}| + \nu_{i,:}}{2} \\
&= \frac{1}{2}\left( \sum_{i \in \mathcal{I}} \ell_i |\nu_{i,:}| + \sum_{i \in \mathcal{I}} \ell_i \nu_{i,:} \right) \\
&= \frac{1}{2}\left( \sum_{i \in \mathcal{I}} \ell_i |g_j(I)_i| + \sum_{i \in \mathcal{I}} \ell_i g_j(I)_i \right)
\end{aligned}
\tag{50}
$$

Note that since $g_j$ is just a linear function that does a forward pass through the network, for any matrix $A, B$,

$$
A g_j(B) = ABD_j W_j^T \ldots D_n W_n^T = g_j(AB).
$$

So we can take the multiplication by scaling terms $\ell$ to be an operation on the input to the network (note that we assume $\ell_i < 0$, which is true for all $i \in \mathcal{I}$)

$$
\sum_{i \in \mathcal{I}} \ell_i[\nu_{i,:}]_+ = \frac{1}{2}\left( -\sum_{i \in \mathcal{I}} |g_j(\text{diag}(\ell))_i| + \sum_{i \in \mathcal{I}} g_j(\text{diag}(\ell))_i \right)
\tag{51}
$$

Similarly, we can view the summation over the index set $\mathcal{I}$ as a summation after multiplying by an indicator matrix $1_{\mathcal{I}}$ which zeros out the ignored rows. Since this is also linear, we can move it to be an operation on the input to the network.

$$
\sum_{i \in \mathcal{I}} \ell_i[\nu_{i,:}]_+ = \frac{1}{2}\left( -\sum_{i} |g_j(1_{\mathcal{I}} \text{diag}(\ell))_i| + \sum_{i} g_j(1_{\mathcal{I}} \text{diag}(\ell))_i \right)
\tag{52}
$$

Let the linear, preprocessing operation be $h(X) = X 1_{\mathcal{I}} \text{diag}(\ell)$ so

$$
h(I) = 1_{\mathcal{I}} \text{diag}(\ell).
$$

Then, we can observe that the two terms are simply an $\ell_{1,:}$ operation and a summation of the network output after applying $g_j$ to $h(I)$ (where in the latter case, since everything is linear we can take the summation inside both $g$ and $h$ to make it $g_j(h(1^T))$):

$$
\sum_{i \in \mathcal{I}} \ell_i[\nu_{i,:}]_+ = \frac{1}{2}\left( -\|g_j(h(I))\|_{1,:} + g_j(h(1^T)) \right)
\tag{53}
$$

The latter term is cheap to compute, since we only pass a single vector. We can approximate the first term using the median estimator on the compound operations $g \circ h$ for a Cauchy random matrix $R$:

$$
\sum_{i \in \mathcal{I}} \ell_i[\nu_{i,:}]_+ \approx \frac{1}{2}\left( -\text{median}(|g_j(h(R))|) + g_j(h(1^T)) \right)
\tag{54}
$$

The end result is that this term can be estimated by generating a Cauchy random matrix, scaling its terms by $\ell$ and zeroing out columns in $\mathcal{I}$, then passing it through the network and taking the median. $h(R)$ can be computed for each layer lower bounds $\ell$, and cached to be computed for the next layer, similar to the non-approximate case.

## E  High probability bounds

In this section, we derive high probability certificates for robustness against adversarial examples. Recall that the original certificate is of the form

$$
J(g(c, \alpha)) < 0,
$$

so if this holds we are guaranteed that the example cannot be adversarial. What we will show is an equivalent high probability statement: for $\delta > 0$, with probability at least $(1 - \delta)$,

$$
J(g(c, \alpha)) \leq \tilde{J}(g(c, \alpha))
$$

where $\tilde{J}$ is equivalent to the original $J$ but using a high probability $\ell_1$ upper bound. Then, if $\tilde{J}(g(c, \alpha)) < 0$ then with high probability we have a certificate.

### E.1 High probability bounds using the geometric estimator

While the median estimator is a good heuristic for training, it is still only an estimate of the bound. At test time, it is possible to create a provable bound that holds with high probability, which may be desired if computing the exact bound is computationally impossible.

In this section, we derive high probability certificates for robustness against adversarial examples. Recall that the original certificate is of the form

$$J(g(c, \alpha)) < 0,$$

so if this holds we are guaranteed that the example cannot be adversarial. What we will show is an equivalent high probability statement: for $\delta > 0$, with probability at least $(1 - \delta)$,

$$J(g(c, \alpha)) \leq \tilde{J}(g(c, \alpha))$$

where $\tilde{J}$ is equivalent to the original $J$ but using a high probability upper bound on the $\ell_1$ norm. Then, if $\tilde{J}(g(c, \alpha)) < 0$ then with high probability we have a certificate.

### E.2 Tail bounds for the geometric estimator

From Li et al. [2007], the authors also provide a geometric mean estimator which comes with high probability tail bounds. The geometric estimator is

$$\|\hat{\nu}_1\|_{1,j} \approx \prod_{i=1}^{k} |g(R)_{i,j}|^{1/k}$$

and the relevant lower tail bound on the $\ell_1$ norm is

$$P\left( \frac{1}{1-\epsilon} \prod_{i=1}^{k} |g(R)_{i,j}|^{1/k} \leq \|\hat{\nu}_1\|_{1,j} \right) \leq \exp\left( -k\frac{\epsilon^2}{G_{L,gm}} \right) \tag{55}$$

where

$$G_{L,gm} = \frac{\epsilon^2}{\left( -\frac{1}{2} \log\left( 1 + \left(\frac{2}{\pi} \log(1-\epsilon)\right)^2 \right) + \frac{2}{\pi} \tan^{-1}\left(\frac{2}{\pi} \log(1-\epsilon)\right) \log(1-\epsilon) \right)}$$

Thus, if $\exp\left( -k\frac{\epsilon^2}{G_{L,gm}} \right) \leq \delta$, then with probability $1 - \delta$ we have that

$$\|\hat{\nu}_1\|_{1,j} \leq \frac{1}{1-\epsilon} \prod_{i=1}^{k} |g(R)_{i,j}|^{1/k} = \text{geo}(R)$$

which is a high probability upper bound on the $\ell_1$ norm.

### E.3 Upper bound on $J(g(c, \alpha))$

In order to upper bound $J(g(c, \alpha))$, we must apply the $\ell_1$ upper bound for *every* $\ell_1$ term. Let $n_1, \ldots, n_k$ denote the number of units in each layer of a $k$ layer neural network, then we enumerate all estimations as follows:

1. The $\ell_1$ norm computed at each intermediary layer when computing iterative bounds. This results in $n_2 + \cdots + n_{k-1}$ estimations.

2. The $\sum_{j \in \mathcal{I}_i} \ell_{i,j}[\nu_{i,j}]_+$ term for each $i = 2, \ldots, k-1$, computed at each intermediary layer when computing the bounds. This results in $n_3 + 2n_4 + \cdots + (k-3)n_{k-1}$.

In total, this is $n_2 + 2n_3 + \cdots + (k-2)n_{k-1} = N$ total estimations. In order to say that *all* of these estimates hold with probability $1 - \delta$, we can do the following: we bound each estimate in Equation 55 with probability $\delta/N$, and use the union bound over all $N$ estimates. We can then conclude that with probability at most $\delta$, any estimate is not an upper bound, and so with probability $1 - \delta$ we have a proper upper bound.

## E.4 Achieving $\delta/N$ tail probability

There is a problem here: if $\delta/N$ is small, then $\epsilon$ becomes large, and the bound gets worse. In fact, since $\epsilon < 1$, when $k$ is fixed, there's actually a lower limit to how small $\delta/N$ can be.

To overcome this problem, we take multiple samples to reduce the probability. Specifically, instead of directly using the geometric estimator, we use the maximum over multiple geometric estimators

$$\text{maxgeo}(R_1, \ldots, R_m) = \max(\text{geo}(R_1), \ldots, \text{geo}(R_m)),$$

where $R_i$ are independent Cauchy random matrices. If each one has a tail probability of $\delta$, then the maximum has a tail probability of $\delta^m$, which allows us to get arbitrarily small tail probabilities at a rate exponential in $m$.

## E.5 High probability tail bounds for network certificates

Putting this altogether, let $\delta > 0$, let $N > 0$ be the number of estimates needed to calculate a certificate, and let $m$ be the number of geometric estimators to take a maximum over. Then with probability $(1 - \delta)$, if we bound the tail probability for each geometric estimate with $\hat{\delta} = \left(\frac{\delta}{N}\right)^{1/m}$, then we have an upper bound on the certificate.

**MNIST example**  As an example, suppose we use the MNIST network from Wong and Kolter [2017]. Then, let $\delta = 0.01$, $m = 10$, and note that $N = 6572$. Then, $\hat{\delta} = 0.26$, which we can achieve by using $k = 200$ and $\epsilon = 0.22$.

Figure 6: Histograms of the relative error of the median estimator for 10 (top), 50 (middle), and 100 (bottom) projections, for a (left) random and (right) robustly trained convolutional layer.

Figure 7: Timing (top) and memory in MB (bottom) plots for a single 3 by 3 convolutional layer to evaluate 10 MNIST sized examples with minibatch size 1, averaged over 10 runs. The number of hidden units is varied by increasing the number of filters. On a single Titan X, the exact method runs out of memory at 52,800 hidden units, whereas the random projections scales linearly at a slope of $2.26 \times 10^{-7}$ seconds per hidden unit, up to 0.96 seconds for 4,202,240 hidden units.

### E.6 Estimation quality and speedup

In this section, we discuss the empirical quality and speedup of the median estimator for $\ell_1$ estimation (for a more theoretical understanding, we direct the reader to Li et al. [2007]). In Figure 6, we plot the relative error of the median estimator for varying dimensions on both an untrained and a trained convolutional layer, and see that regardless of whether the model is trained or not, the distribution of the estimate is normally distributed with decreasing variance for larger projections, and without degenerate cases. This matches the theoretical results derived in Li et al. [2007].

In Figure 7, we benchmark the time and memory usage on a convolutional MNIST example to demonstrate the performance improvements. While the exact bound takes time and memory that is quadratic in the number of hidden units, the median estimator is instead linear, allowing it to scale up to millions of hidden units whereas the exact bound runs out of memory out at 50,280 hidden units.

## F AutoDual

In this section, we describe our generalization of the bounds computation algorithm from [Wong and Kolter, 2017] to general networks using dual layers, which we call AutoDual.

**Efficient construction of the dual network via linear dual operators** The conjugate form, and consequently the dual layer, for certain activations requires knowing lower and upper bounds for the pre-activations, as was done for ReLU activations in Algorithm 1 of Wong and Kolter [2017]. While the bound in Equation 7 can be immediately used to compute all the bounds on intermediate nodes of the network one layer at a time, this requires performing a backwards pass through the dual network whenever we need to compute the bounds. However, if the operators $g_{ij}$ of the dual layers are all affine operators $g_{ij}(\nu_{i+1}) = A_{ij}^T \nu_{i+1}$ for some affine operator $A_{ij}$, we can apply a generalization of the lower and upper bound computation found in Wong and Kolter [2017] to compute all lower and upper bounds, and consequently the dual layers, of the entire network with a single forward pass in a layer-by-layer fashion. With the lower and upper bounds, we can also use the same algorithm to automatically construct the dual network. The resulting algorithm, which we call AutoDual, is described in Algorithm 3.

In practice, we can perform several layer-specific enhancements on top of this algorithm. First, many of the $A_{ji}$ operators will not exist simply because most architectures (with a few exceptions) don't have a large number of skip connections, so these become no ops and can be ignored. Second, we can lazily skip the computation of layer-wise bounds until necessary, e.g. for constructing the dual layer of ReLU activations. Third, since many of the functions $h_j$ in the dual layers are functions of $B^T \nu_i$ for some matrix $B$ and some $i \geq j$, we can initialize $\nu_i^{(i)}$ with $B$ instead of the identity matrix, typically passing a much smaller matrix through the dual network (in many cases, $B$ is a single vector).

---

**Algorithm 3** Autodual: computing the bounds and dual of a general network

---

**input:** Network operations $f_{ij}$, data point $x$, ball size $\epsilon$
*// initialization*
$\nu_1^{(1)} := I$
$\ell_2 := x - \epsilon$
$u_2 := x + \epsilon$
**for** $i = 2, \ldots, k-1$ **do**
   *// initialize new dual layer*
   Create dual layer operators $A_{ji}$ and $h_i$ from $f_{ji}, \ell_j$ and $u_j$ for all $j \leq i$
   $\nu_i^{(i)} := I.$
   *// update all dual variables*
   **for** $j = 1, \ldots, i-1$ **do**
      $\nu_j^{(i)} := \sum_{k=1}^{j-1} A_{ki}\nu_j^{(k)}$
   **end for**
   *// compute new bounds*
   $\ell_{i+1} := x^T\nu_1^{(i)} - \epsilon\|\nu_1^{(i)}\|_: + \sum_{j=1}^{i} h_j(\nu_j^{(i)}, \ldots, \nu_i^{(i)})$
   $u_{i+1} := x^T\nu_1^{(i)} + \epsilon\|\nu_1^{(i)}\|_: - \sum_{j=1}^{i} h_j(-\nu_j^{(i)}, \ldots, -\nu_i^{(i)})$
   *// $\|\cdot\|_:$ for a matrix here denotes the norm of all rows*
**end for**
**output:** bounds $\{\ell_i, u_i\}_{i=2}^{k}$, dual layer operators $A_{jk}, h_i$

---

# G  Experiments

In this section, we provide more details on the experimental setup, as well as more extensive experiments on the effect of model width and model depth on the performance that were not mentioned above.

We use a parameter $k$ to control the width and depth of the architectures used in the following experiments. The Wide($k$) networks have two convolutional layers of $4 \times k$ and $8 \times k$ filters followed by a $128 \times k$ fully connected layer. The Deep($k$) networks have $k$ convolutional filters with 8 filters followed by $k$ convolutional filters with 16 filters.

**Downsampling**  Similar to prior work, in all of our models we use strided convolutional layers with 4 by 4 kernels to downsample. When downsampling is not needed, we use 3 by 3 kernels without striding.

## G.1  MNIST

**Experimental setup**  For all MNIST experiments, we use the Adam optimizer with a learning rate of 0.001 with a batch size of 50. We schedule $\epsilon$ starting from 0.01 to the desired value over the first 20 epochs, after which we decay the learning rate by a factor of 0.5 every 10 epochs for a total of 60 epochs.

**Model width and depth**  We find that increasing the capacity of the model by simply making the network deeper and wider on MNIST is able boost performance. However, when the model becomes overly wide, the test robust error performance begins to degrade due to overfitting. These results are shown in Table 3.

## G.2  CIFAR10

**Experimental setup**  For all CIFAR10 experiments, we use the SGD optimizer with a learning rate of 0.05 with a batch size of 50. We schedule $\epsilon$ starting from 0.001 to the desired value over the first 20 epochs, after which we decay the learning rate by a factor of 0.5 every 10 epochs for a total of 60 epochs.

Table 3: Results on different widths and depths for MNIST

| Dataset | Model | Epsilon | Robust error | Error |
|---|---|---|---|---|
| MNIST | Wide(1) | 0.1 | 6.51% | 2.27% |
| MNIST | Wide(2) | 0.1 | 5.46% | 1.55% |
| MNIST | Wide(4) | 0.1 | 4.94% | 1.33% |
| MNIST | Wide(8) | 0.1 | 4.79% | 1.32% |
| MNIST | Wide(16) | 0.1 | 5.27% | 1.36% |
| MNIST | Deep(1) | 0.1 | 5.28% | 1.78% |
| MNIST | Deep(2) | 0.1 | 4.37% | 1.28% |
| MNIST | Deep(3) | 0.1 | 4.20% | 1.15% |

Table 4: Results on MNIST, and CIFAR10 with small networks, large networks, residual networks, and cascaded variants for $\ell_2$ perturbations.

| Dataset | Model | Epsilon | Single model error | | Cascade error | |
|---|---|---|---|---|---|---|
| | | | Robust | Standard | Robust | Standard |
| MNIST | Small, Exact | 1.58 | 56.48% | **11.86%** | **24.42%** | **19.57%** |
| MNIST | Small | 1.58 | 56.32% | 13.11% | 25.34% | 20.93% |
| MNIST | Large | 1.58 | **55.47%** | 11.88% | 26.16% | 24.97% |
| CIFAR10 | Small | 36/255 | 53.73% | 44.72% | 50.13% | 48.64% |
| CIFAR10 | Large | 36/255 | 49.40% | 40.24% | **41.36%** | **41.16%** |
| CIFAR10 | Resnet | 36/255 | **48.04%** | **38.80%** | 41.44% | 41.28% |

# H   Results for $\ell_2$ perturbations

We run similar experiments for $\ell_2$ perturbations on the input instead of $\ell_\infty$ perturbations, which amounts to replacing the $\ell_1$ norm in the objective with the $\ell_2$ norm. This can be equivalently scaled using random normal projections [Vempala, 2005] instead of random Cauchy projections. We use the same network architectures as before, and pick $\epsilon_2$ such that the volume of an $\ell_2$ ball with radius $\epsilon_2$ is approximately the same as the volume of an $\ell_\infty$ ball with radius $\epsilon_\infty$. A simple conversion (an overapproximation within a constant factor) is:

$$\epsilon_2 = \sqrt{\frac{d}{\pi}}\epsilon_\infty.$$

For MNIST, we take an equivalent volume to $\epsilon_\infty = 0.1$. This ends up being $\epsilon_2 = 1.58$, and note that within the dataset, the minimum $\ell_2$ distance between any two digits is at least 3.24, so $\epsilon_2$ is roughly half of the minimum distance between any two digits. For CIFAR we take an equivalent volume to $\epsilon_\infty = 2/255$, which ends up being $\epsilon_2 = 36/255$.

The results for the complete suite of experiments are in Table 4, and we get similar trends in robustness for larger and cascaded models to that of $\ell_\infty$ perturbations.