[Reviews · NeurIPS 2018]

Reviewer 1



This paper proposes 3 improvements of [1] for training provably robust neural networks. The 1st is an algorithms for automatically constructing a "dual network" for calculating the dual bound. The 2nd uses random projection to reduce computation. The 3rd introduces the cascade trick to further reduce robust error at the price of increased clean test error. The paper is clearly written. These improvements over [1] is novel and significant, allowing the training of larger provably robust networks with more versatile structures. In addition, I need some clarifications for Figure 3. The cascade models are trained sequentially? is that correct? say, Cascade 2 is trained after Cascade 1 is completely trained? So the epoch number in Figure 3 (train) doesn't indicate concurrency? For the testing curves, when we look at Cascade 3 epoch 1, the Cascade 3 model just started training? and Cascade 4 and 5 hasn't been trained? I recommend acceptance for this paper. [1] Eric Wong and J Zico Kolter. Provable defenses against adversarial examples via the convex outer adversarial polytope. arXiv preprint arXiv:1711.00851, 2017.

Reviewer 2



__ Post-rebuttal feedback: I thank the authors for the clarification. Based on the rebuttal letter, in the final version I'd suggest emphasizing the provable defense is guaranteed in probabilistic sense. Even though I agree in test time the geometric estimator is not necessary, what you indeed certified are training data, instead of test data. This is a nice piece of work and I enjoy reading it. __ This paper primarily builds upon the seminal work in [Wong-Kolter’17] to show improved and generalized verification and model robustness evaluation in three aspects: (I) extension to general networks (e.g., those with skip connections) and non-ReLU activation functions; (II) make the proposed verification method scalable via a nonlinear random projection technique, exhibiting linear computation complexity in the input dimension and the number of hidden units; (III) Verifiable bound improvement using model cascades. In my opinion, this work has made important contributions in norm-bounded robustness verification by proposing a scalable and more generic toolkit for robustness certification. The autodual framework is both theoretically grounded and algorithmically efficient. However, I also have two major concerns about this work: (I) the proposed nonlinear random projection leads to an estimated (i.e., probabilistic) lower bound of the minimum distortion towards misclassification, which is a soft robustness certification and does not follow the mainstream definition of deterministic lower bound; (II) Since this method yields an estimated lower bound, it then lacks performance comparison to existing bound estimation methods. Strength: 1. This paper is well-written and it proposes a theoretically grounded and practically important solution to support scalable robustness verification of neural networks with different network architectures (e.g., skip connections) and non-ReLU activation functions. 2. To the best of my knowledge, this work is the first verification method that can scale up to more than 10K hidden units within a reasonable computation time. Weakness: 1. An estimated lower bound (on minimum distortion for misclassification) is not a lower bound: I was very concerned that the current write-up may give a false sense of scalable “deterministic” defense. In order to speed up the computation, the use of the random projection makes the lower bound an estimation rather than a hard certificate. However, the notion of statistical lower bound (that means with some probability the result is not a lower bound) actually differs from the mainstream definition of verification – which needs to be deterministically verified, such as the definition in [Wong-Kolter’17]. Although the authors have discussed in the appendix on how to convert this probabilistic bound to the associated lower bound for certification, I am not quite convinced that one should call the proposed method a “provable defense”. I suggest modifying the title to be "Scaling adversarial defenses with statistical guarantees". The use of "provable" here is very vague and may be messed up with deterministic robustness certification. In the introduction, the authors should also highlight that the notion of robustness certification in this paper is statistical rather than deterministic. 2. Lack of performance comparison to existing methods: Since the proposed scalable approach yields a statistical estimate of a lower bound, there are already existing works that use extreme value theory to estimate the a lower bound, which is scalable to deep ImageNet classifiers and empirically aligned with the results of strong attacks (i.e., tight upper bounds). Due to the same spirit of efficient statistical estimation of a lower bound, I suggest the authors to compare with [R1] by fixing the same network and setting the \epsilon in Table 2 to be the same as the estimated lower bound in [R1] (by tuning ), and compare the resulting robust errors. 3. Extension beyond ReLU: One of the main contributions claimed in this work is the capability of certifying robustness for networks with non-ReLU activation functions. However, the experimental results are all based on ReLU networks and in the manuscript the authors merely refer readers to [Dvijotham et al., 2018]. I suggest including additional descriptions on the detailed procedures of incorporating the results in [Dvijotham et al., 2018] into the autodual framework and provide empirical results on non-ReLU networks for proper justification. Summary: This paper has made important contributions in scaling up the robustness certification method. However, the proposed method trades in the commonly used deterministic robustness guarantee to a probabilistic one (the latter is not a mainstream definition of certified robustness under norm-bounded perturbation attacks), and lacks performance comparison to existing methods that also use statistical estimates of lower bounds. Therefore, my main concerns are the credibility of probabilistic robustness guarantee and lacking performance comparison to other statistical robustness certification approaches. [R1] Evaluating the Robustness of Neural Networks: An Extreme Value Theory Approach

Reviewer 3



Main idea: This paper studies the adversarial defense for the deep learning models. Compare with the traditional deep learning defense model that suffers from the problems of constrained architecture and not linearly scalable. This paper proposes a linearly yet adaptive defense model that shows good performance compare with the state-of-the-art frameworks. Strengths: The written is good and logic is easy to follow. Introduction, as well as the related work, are thorough and complete. The derivation of the dual network is novel and intuitively more efficient in computation. The further bound analysis using random project with mean estimator can further speed up the calculation and cascading ensemble could reduce the potential bias The code with the submission serves good as a supplementary for the understanding of general audiences. Weakness: One of the claimed contributions of this framework is improved scalability. It would be reasonable to theoretical analysis what is the converging speed improvement compared with the former approach. The corresponding experiments regarding this linear scalability are not conducted and this should be added in the final version. How good is the median estimation in Theorem 2, is there any degenerate cases caused by this approximation? The corresponding analysis is missing. Regarding experimental results, the performance in terms of the error rate looks promising. However, the plots shown in Fig 2 need the explanation to highlights the advantage of using random projections. It looks like the curves are struggling with converging at the early stages and various projection dimensions are not that different in terms of convergence. The appendix has quite a few typos that need to be fixed. e.g. missing references, inconsistent notations (regarding dual norm), etc Quality: Overall, the writing quality is very good, the main idea is clearly presented. Clarity: The notations are simple and well-organized. The presentation has a few issues, but the appendix could be improved by fixing typos. Originality: It is an incremental work with very considerable novelty. Significance: Theoretical proofs are novel, experiments still have space for improvement.